# NeuroSkin^®^: AI-Driven Wearable Functional Electrical Stimulation for Post-Stroke Gait Recovery—A Multicenter Feasibility Study

**DOI:** 10.3390/s25185614

**Published:** 2025-09-09

**Authors:** Amine Metani, Lana Popović-Maneski, Perrine Seguin, Julie Di Marco

**Affiliations:** 1Kurage, 69007 Lyon, France; lana.popovic@kurage.fr (L.P.-M.); perrine.seguin@kurage.fr (P.S.); 2Laboratoire de Physique, École Normale Supérieure (ENS) de Lyon, Université Claude Bernard Lyon 1, Université de Lyon (UdL), Centre National de la Recherche Scientifique (CNRS), 69007 Lyon, France; 3Institute of Technical Sciences of SASA, Knez Mihailova 35/IV, 11000 Belgrade, Serbia; 4Centre de Soins de Suite et de Réadaptation du Val Rosay, 69370 Saint-Didier au Mont d’Or, France; julie.dimarco@ugecam.assurance-maladie.fr

**Keywords:** Functional Electrical Stimulation (FES), gait rehabilitation, stroke, wearable neurotechnology, Artificial Intelligence (AI), personalized rehabilitation, clinical feasibility, sensor-based therapy

## Abstract

**Highlights:**

**What are the main findings?**
The NeuroSkin^®^ system was safely implemented in seven rehabilitation centers and achieved excellent usability (mean SUS score: 84.6), with no adverse events reported.Patients showed statistically significant improvements in gait speed, endurance, balance, and ambulation level following NeuroSkin^®^-assisted therapy.

**What is the implication of the main finding?**
AI-driven wearable FES can be integrated into routine stroke rehabilitation with minimal training, enabling real-time, multi-muscle stimulation adapted to each patient.These findings support the feasibility of deploying personalized, sensor-based FES in clinical practice and lay the groundwork for future controlled trials assessing its efficacy.

**Abstract:**

(1) Background: Functional Electrical Stimulation (FES) is a recognized method for post-stroke gait rehabilitation but remains underutilized due to workflow complexity and the need for manual configuration. NeuroSkin^®^, a wearable FES system integrating AI-driven stimulation and sensor-based gait monitoring, was developed to streamline clinical use by automating phase-specific, multi-muscle stimulation. (2) Methods: This retrospective multicenter feasibility study evaluated the integration of NeuroSkin^®^ into routine inpatient rehabilitation. Fifteen subacute stroke patients across seven centers underwent 10 to 20 FES-assisted gait training sessions. Standardized assessments (10MWT, 6MWT, TUG, NFAC) were performed pre- and post-intervention. Therapists completed the System Usability Scale (SUS) questionnaire. (3) Results: All outcomes showed statistically significant improvement: walking speed and endurance increased by 70% and 171% respectively, TUG time decreased by 39%, and ambulation level improved by three NFAC categories. No adverse events were reported, and usability was rated as excellent (mean SUS score: 84.6). (4) Conclusions: NeuroSkin^®^ was safely and effectively implemented in diverse clinical settings, demonstrating strong usability and promising functional benefits. These findings support the need for prospective controlled trials to confirm its clinical efficacy and broader applicability in stroke rehabilitation.

## 1. Introduction

Stroke remains the leading cause of acquired neurological disability worldwide, with gait recovery being a central focus of post-stroke rehabilitation efforts [1,2]. Despite advances in therapy, a substantial proportion of survivors retain walking deficits that reduce autonomy and quality of life.

Functional Electrical Stimulation (FES) has been used for decades to improve motor function in individuals with upper motor neuron lesions, including post-stroke patients [3,4,5]. When applied during walking, FES can enhance muscle recruitment, improve gait speed and symmetry, and support neuromuscular plasticity [6,7,8]. Recent systematic reviews and meta-analyses strengthen this evidence in post-stroke populations, with reported significant improvements in gait speed, cadence, step length, ankle dorsiflexion, and 6MWT distance when NMES was integrated into rehabilitation programs [9]; enhanced biomechanical gait parameters, including COP symmetry and joint kinematics [10]; and significant improvements in walking speed and functional activity levels compared with conventional rehabilitation [11]. Together, these findings provide strong support for the use of FES in post-stroke gait recovery.

However, widespread adoption in clinical practice remains limited. Key barriers include the complexity of device setup, the need for manual electrode placement, and the requirement for advanced expertise to adjust stimulation parameters [12,13,14]. Most commercially available FES systems are either overly simplistic (e.g., single- or dual-channel devices addressing isolated foot drop) or highly technical, requiring manual electrode placement, individual calibration, and therapist training. These workflow complexities and steep learning curves have contributed to FES being underused in routine post-stroke rehabilitation [11,15,16,17,18]

Recent developments in wearable technologies and Artificial Intelligence (AI) offer an opportunity to overcome these challenges by automating key elements of FES therapy. NeuroSkin^®^ (Kurage, France) is a novel textile-based FES system that integrates embedded dry electrodes, real-time gait analysis using inertial and force sensors, and AI-driven stimulation timing. Unlike traditional FES systems, NeuroSkin^®^ does not require manual electrode placement or parameter tuning and can deliver multi-muscle, phase-specific stimulation with minimal therapist input. To date, published literature on NeuroSkin^®^ is limited to its technical characterization [19], and the present study provides one of the first multicenter clinical evaluations of its feasibility and usability.

The aim of this retrospective multicenter feasibility study was to evaluate the integration of NeuroSkin^®^ into real-world post-stroke rehabilitation settings. It does not provide evidence of efficacy, only feasibility and preliminary effectiveness. We assessed its safety, usability, and potential clinical impact on walking function as a preliminary step toward larger-scale efficacy trials. It is the first multicenter clinical feasibility study of an AI-driven, textile-integrated, multi-channel FES system for post-stroke gait rehabilitation, evaluated in real-world inpatient settings across multiple independent centers.

## 2. Materials and Methods

### 2.1. Study Design

This multicenter retrospective feasibility study aimed to assess the integration of the NeuroSkin^®^ system (Kurage, Lyon, France)—an AI-driven wearable FES device—into routine inpatient stroke rehabilitation and describe associated changes in gait function. The sessions were performed in seven rehabilitation centers in France, Luxembourg, and Italy, with inpatients at the early subacute phase of rehabilitation, under the supervision of local physiotherapists. In addition to the NeuroSkin^®^ intervention, all patients received 1 to 2 h of standard physiotherapy per day, five days a week, in accordance with usual care protocols at each center. Standard walking assessments were conducted before and after FES device usage as part of the usual clinical routine. Each center’s physiotherapist completed the System Usability Scale (SUS) questionnaire [20] post-program, while patients were invited to complete a basic satisfaction survey.

Ethics compliance and data collection and analysis were carried out in accordance with the MR004 reference methodology, which governs the processing of personal health data for non-interventional retrospective research in France. Under MR004, such studies require declaration to the French Data Protection Authority (CNIL). In addition, all patients or legal guardians provided informed consent for treatment and use of anonymized data.

The overall clinical workflow using the NeuroSkin^®^ system is summarized in Figure 1, from initial training of the physiotherapists to outcome assessment.

Inclusion criteria:First-ever supratentorial ischemic or hemorrhagic stroke;Subacute stage (≤6 months post-stroke);Hemiparetic gait requiring assistance with sufficient motor skills to walk with help from a single person or technical aids (0 < NFAC < 5—New Functional Ambulation Category [21]);Medically stable and able to participate in therapy;Responsive to FES.

Exclusion criteria:Orthopedic or cardiorespiratory contraindications to walking;Implanted electrical devices;Cognitive or communication impairments precluding evaluation;Not responsive to FES.

Subacute stroke patients were specifically targeted for this feasibility study, which was motivated by the following factors. Acute patients were excluded because few are able to walk safely at this stage, and many remain medically unstable, limiting the feasibility of FES-assisted gait training. From a neurophysiological standpoint, spontaneous neuroplasticity and functional recovery potential are highest during the acute and early subacute phases—typically within the first three months after stroke—making the subacute period an optimal window for evaluating rehabilitation interventions. Clinical data support that a therapeutic sensitivity window of approximately 60–90 days post-stroke is associated with heightened treatment responsiveness, whereas interventions in the chronic phase (>6 months) exhibit markedly reduced efficacy [22]. Clinically, gait training is typically prioritized during this stage, as most rehabilitation programs intensify mobility-focused therapy during early recovery. Chronic patients were also excluded, as functional recovery generally plateaus beyond this period. Finally, from a pragmatic standpoint, the participating rehabilitation centers primarily admitted subacute inpatients; focusing on this population reduced variability and improved interpretability of findings, given the limited sample size.

Cognitive or communication impairments precluding evaluation were assessed based on the treating therapists’ clinical judgment. Before enrollment, patients were screened for responsiveness to electrical stimulation by applying a brief test stimulation with any available TENS (Transcutaneous Electrical Nerve Stimulation) device on the quadriceps muscle; only patients showing a visible contraction sufficient to extend the leg against gravity were included.

### 2.2. Intervention

Each therapy session lasted between 30 and 45 min and consisted of FES-assisted gait training, where only the paretic leg was stimulated, supervised by a local physiotherapist trained in using the NeuroSkin^®^ system. A minimum of 10 FES gait therapy sessions were performed (maximum: 20; average: 16.1 ± 4.27), depending on local clinical protocols, individual rehabilitation goals, and length of inpatient stay. The intervention followed a personalized and adaptive approach rather than a fixed standardized protocol, to reflect real-world clinical practice.

-Personalization: Before the first session, a brief personalization phase was conducted, during which the patient walked approximately 20 steps without stimulation. Gait data from this recording were used to refine the pre-trained gait model and improve the accuracy of gait phase detection for stimulation control.-Calibration of stimulation intensities: For each of the six targeted muscle groups (gluteus maximus, quadriceps, hamstrings, tibialis anterior, fibularis, gastrocnemius), the maximum tolerable stimulation intensity was determined individually at baseline. These limits could later be adjusted as needed during sessions to account for habituation, fatigue, or recovery progress.-Gait training activities: Most sessions involved repeated overground walking at a comfortable pace along a flat indoor corridor, using technical aids (e.g., canes, walkers) if required. Therapists focused on correcting gait deficits common after stroke, such as drop foot, knee hyperextension, or equinovarus deformity, by tuning stimulation timing and intensity in real time.-Session structure and progression: Sessions typically alternated between short walking bouts and brief rest periods depending on patient fatigue and tolerance. Progression across sessions was individualized and could involve increased walking distance, higher stimulation intensities, or focusing on additional gait phases and muscle groups as recovery advanced.-Real-time adjustments: During each session, therapists used the NeuroSkin^®^ tablet interface and the remote control to modify stimulation parameters on the fly, allowing them to adapt to patient-specific gait deficits and comfort levels dynamically.

The physiotherapists’ training consisted of a 2 h course about the basics of FES, followed by a 2 h hands-on workshop on how to use the NeuroSkin^®^ system, where they would try the device on themselves. The workshop’s objective was to teach physiotherapists how to adapt the stimulation parameters, especially intensities and timings, in order to help correct specific gait deficits often encountered in the post-stroke population, like the abovementioned. Given that the device is designed for single-operator use, each center assigned one therapist, yielding a total of seven therapists across all sites.

As previously stated, session protocols were not harmonized: given the wide diversity of post-stroke patients’ specificities, the physiotherapists were instructed to tailor the sessions to the needs of the patients, relying on their own expertise. While this introduces inter-site variability, it reflects real-world implementation and reinforces the generalizability of the findings. Physiotherapists were asked to train the participant at the maximum comfortable intensity, in order to maximize the muscle tone benefits induced by the electrical stimulation, and to look out for specific gait deficits that could be corrected using FES, as trained during the workshop. This pragmatic design aimed to mirror real-world clinical conditions while enabling therapists to tailor the intervention to each patient’s functional abilities and rehabilitation goals.

### 2.3. Outcome Measures and Statistical Analysis

Clinical outcomes were assessed at baseline and after the intervention using four standardized measures commonly used in stroke rehabilitation research:10-Meter Walk Test (10MWT): evaluates gait speed by timing how long it takes to walk 10 m at a comfortable pace [23];6-Minute Walk Test (6MWT): measures walking endurance by recording the total distance walked in six minutes [24];Timed Up and Go (TUG): assesses functional mobility and dynamic balance by timing how long it takes to stand up from a chair, walk 3 m, turn, return, and sit down [25];New Functional Ambulation Classification (NFAC): categorizes the level of walking autonomy on a nine-point scale, ranging from non-functional ambulation to independent walking [21].

These outcome measures were part of the standardized clinical routine across participating centers and were therefore used for the retrospective analysis. They are widely validated in post-stroke populations, capture complementary dimensions of gait recovery (speed, endurance, balance, and autonomy), and facilitate comparison with existing literature. All assessments were performed by the same physiotherapists who administered the FES-assisted gait training sessions. No independent or blinded assessors were involved, which is consistent with the pragmatic, real-world design of this feasibility study. In addition, baseline descriptive variables (age, sex, and time since stroke) were summarized using descriptive statistics. Continuous variables are reported as mean ± standard deviation, and categorical variables as counts.

For each outcome measure, analyses were conducted using only patients with both baseline and post-intervention data available. No imputation was performed for missing data. Statistical analyses were conducted to compare pre- and post-intervention outcomes. Normality of the differences between paired measurements was assessed using the Shapiro–Wilk test. When the normality assumption was met (*p* > 0.05), a two-tailed paired *t*-test was used; otherwise, the Wilcoxon signed-rank test was applied. For parametric tests, significance was reported using *p*-values and effect sizes expressed as Cohen’s d; for non-parametric tests, effect sizes were calculated as the r-value (Z-score divided by the square root of the sample size). A *p*-value below 0.05 was considered statistically significant.

### 2.4. Usability and Satisfaction

Therapists evaluated usability using the System Usability Scale (SUS) [20], a validated 10-item questionnaire widely used to assess perceived usability of interactive systems. Each item is rated on a 5-point Likert scale from 1 (“strongly disagree”) to 5 (“strongly agree”). Individual item scores are converted to a total score ranging from 0 to 100, where higher scores indicate better perceived usability. According to established guidelines, scores above 68 are considered above average, scores above 80 indicate excellent usability, and scores above 90 represent the best imaginable usability.

In addition, at the end of the intervention, patients were invited to complete a basic satisfaction survey, providing a single self-reported rating of their overall satisfaction with the NeuroSkin^®^-assisted rehabilitation sessions on a numeric scale from 1 (“very dissatisfied”) to 10 (“very satisfied”).

### 2.5. The Neuroskin System

NeuroSkin^®^ is a CE-marked Class IIa medical device, which has been previously described in [19], a preliminary case study that represents one of the earliest clinical implementations of the system. It is a therapeutic device and real-time gait monitoring platform designed to enhance the walking ability of the user by delivering electrical stimulation to the lower limbs at optimal moments (see Figure 2). It integrates the following components:A lower-extremity garment with embedded FES dry electrodes targeting the six following muscle groups: Gluteus Maximus, Quadriceps, Hamstrings, Tibialis Anterior, Fibularis, and Gastrocnemius;A set of sensors: seven Inertial Measurement Units (IMU) placed on the pelvis, upper and lower leg segments, and feet; eight Ground Reaction Force (GRF) sensors integrated into the insoles of the shoes;An AI-driven real-time gait phase detector incorporated into a microcomputer positioned on the back of a vest worn by the patient;A MotiMove (3F-Fit Fabricando Faber, Belgrade, Serbia) electrical stimulator [26];A remote controller used to regulate the overall intensity of stimulation during the sessions;An application allowing therapists to manage individual patient profiles, including stimulation parameters (see Figure 3).

The NeuroSkin^®^ hardware is controlled through a tablet application featuring two main screens: the Global Settings screen and the Expert Settings screen, as displayed in Figure 3. The Global Settings screen allows the operator to oversee multiple patient profiles, including demographic information, paretic side, and maximum allowed intensities for each muscle group. Additionally, it facilitates monitoring of hardware connections and controls session initiation, pause, and termination. The Expert Settings screen provides access to more advanced parameters, such as global frequency for all channels, as well as pulse widths and timings (as a percentage of the gait cycle) for each channel.

The data incoming from the sensors is processed in real time to determine a gait percentage used to trigger electrical pulses based on pre-defined rules tailored to the patient’s gait pattern classification [27]. A general gait model, based on deep learning over a database of 32 healthy and post-stroke hemiplegic walking patterns, is refined following an additional personalization phase, involving the patient walking 20 steps without stimulation. The next section provides more details about the AI model and personalization process.

Before starting the first session, both personalization and calibration procedures have to be performed. Personalization involves using gait data from the patient walking approximately 20 steps without stimulation to partially retrain the model with the individual’s specific gait pattern, in order to improve the accuracy of gait phase classification for stimulation control. Calibration consists of determining the maximum stimulation intensity tolerable by the patient for each muscle. The system is then capped accordingly to prevent stimulation beyond these maximum values. These intensity limits can later be adjusted during any session, depending on the patient’s needs—for example, to compensate for habituation or fatigue.

These two procedures are only required before the first session and do not need to be repeated throughout the rest of the intervention. However, they can be repeated if the therapist deems it necessary to account for substantial progress in the patient’s walking ability.

### 2.6. AI Model and Personalization Procedure

The NeuroSkin^®^ system uses a pre-trained gait-phase detection model based on inertial and pressure sensors. It is based on a deep neural network architecture comprising successive 1D convolutional layers with residual connections [28]. It processes 500 ms windows of real-time gyroscope signals from inertial measurement units (IMUs) to predict the continuous gait phase as a percentage of the stride cycle. The model was trained on a database of gait recordings from 10 healthy and 22 post-stroke hemiparetic individuals. The model outputs the gait phase as a sine–cosine pair representing the phase angle within the gait cycle. Accuracy is measured using the mean squared error (MSE) between predicted and reference values.

To adapt the model to individual patients, a transfer learning procedure is employed. After an initial recording of approximately 20 steps, only the final fully connected layer is retrained using the patient-specific data, while the feature extraction layers remain fixed. This approach strikes a balance between generalization and patient-specific adaptation, allowing the model to capture unique gait features such as asymmetry, low speed, variable terrain adaptation, or fatigue. The personalized model is then compressed and deployed for real-time use.

## 3. Results

Fifteen patients (average age 60.60 ± 12.65; 5 female and 10 male) were included in the study across seven centers. The average time since the stroke was 61.67 ± 28.92 days. Table 1 summarizes their baseline demographic and clinical characteristics.

Individual pre- and post-intervention results for the 10MWT, 6MWT, TUG, and NFAC are summarized in Table 2, along with their normality, significance, and effect size. All outcome measures showed statistically significant improvements.

The observed improvements exceeded published thresholds for clinically meaningful change. The mean increase in gait speed (+0.35 m/s) surpassed both the small (0.06 m/s) and substantial (0.14 m/s) Minimal Clinically Important Changes (MCID) as reported in [29]. Similarly, the median improvement in 6MWT distance (+123.5 m) was far greater than the MCID of 18.6 m identified in [30]. For TUG, although no MCID is available, the observed mean reduction (36.9 s) exceeded both the minimal detectable change (MDC) threshold (2.9 s) and the smallest real difference (SRD) threshold (23%) reported in [30]. No validated MCID exists for the NFAC scale, but the mean gain of three categories strongly suggests meaningful improvements in walking autonomy.

Physiotherapists also completed the SUS questionnaire post-program, as reported in Table 3. The mean SUS score was 84.6 ± 10.1, indicating high perceived usability among trained therapists within this feasibility context.

In addition, patient satisfaction with the intervention was assessed at the end of the program using a single-item numeric rating scale (1 = not satisfied at all; 10 = extremely satisfied), as reported in Table 4. Among the 12 patients who responded (3 non-respondents), the mean satisfaction score was 7.33 ± 2.31. While these preliminary findings suggest a generally positive perception of the therapy, the simplicity of the measure and potential response biases limit their interpretability.

No adverse events or device-related complications were reported during the intervention. However, given the small sample size and short intervention period, these findings provide only an initial safety profile.

## 4. Discussion

This multicenter retrospective feasibility study suggests that NeuroSkin^®^ can be implemented in routine inpatient rehabilitation across diverse clinical settings, with high therapist-reported usability and no reported adverse events. Patients demonstrated improvements in gait speed, endurance, mobility, and ambulation level, all of which exceeded published MCIDs or MDCs for post-stroke populations [29,30], which aligns with prior work supporting FES for post-stroke mobility recovery [6,7,8,9,10,11,12,13,14,15]. These findings indicate that the observed changes are not only statistically significant but also clinically meaningful, supporting the potential of AI-driven, multi-channel FES to augment conventional rehabilitation.

Direct comparisons between NeuroSkin^®^ and existing peroneal nerve FES devices—e.g., WalkAide^®^ (Innovative Neurotronics, Austin, TX, USA), L300^®^ (Ness L300; Bioness Inc., Valencia, CA, USA), ODFS^®^ (ODFS Pace, Odstock Medical Ltd., Salisbury, UK)—should be interpreted cautiously. Most available studies on FES for gait rehabilitation focus on single- or dual-channel devices that target the peroneal nerve to correct isolated foot drop and enhance gait speed in stroke populations [31,32]. In contrast, NeuroSkin^®^ delivers synchronized, phase-specific stimulation to six lower-limb muscle groups, facilitating broader neuromuscular engagement at the cost of increased system complexity—a departure from singular muscle-targeting strategies [33,34]. While this difference may affect feasibility and usability, and despite differences in configuration, safety considerations remain consistent across FES modalities, as the stimulation levels are similar, and adverse events are typically minor [35]. Finally, because this is a retrospective feasibility study without a control group, effectiveness comparisons with other devices would be premature. Future prospective trials should address comparative efficacy, usability, and workflow integration.

Usability was a central outcome in this study. The mean System Usability Scale (SUS) score of 84.6 indicates excellent usability according to established interpretation thresholds [20]. This finding is notable given the greater technical complexity of NeuroSkin^®^ compared with traditional single-channel devices. To our knowledge, few FES studies have reported standardized usability metrics, with one exception being the Fesia Walk device, where therapists reported an SUS score of 85.6 [36]. The comparably high SUS score observed here suggests that the textile design and automated gait phase control of NeuroSkin^®^ may mitigate usability barriers often associated with more sophisticated systems. At the same time, limitations of our usability findings must be acknowledged: only seven therapists were surveyed, all of whom had undergone NeuroSkin^®^-specific training, which may have introduced positive bias. None had prior experience with FES, and their broader clinical experience and digital literacy were not collected. Patient satisfaction was also assessed, though only with a simple 1–10 numeric rating scale, which lacks the depth of validated questionnaires such as the Quebec User Evaluation of Satisfaction with Assistive Technology (QUEST) [37], the Feasibility of Intervention Measure (FIM), or the Acceptability of Intervention Measure (AIM) [38]. Patient-reported outcomes related to comfort or tolerability were not collected, which further constrains interpretation of the usability outcomes. Future studies should therefore integrate standardized patient-reported measures to provide a more comprehensive evaluation of usability and acceptability.

Several methodological limitations must be considered when interpreting these results. First, the small sample size of both patients (*n* = 15) and therapists (*n* = 7) reduces statistical power, limits generalizability, and precludes analyses controlling for confounders such as time since stroke, baseline severity, or number of sessions. Second, the retrospective design and absence of a control group prevent causal inference, and potential biases were introduced because the same therapists who delivered the intervention also performed the outcome assessments. Third, there was considerable heterogeneity in the number of NeuroSkin^®^ sessions (10–20) and variability in conventional therapy content across centers, which might reflect differences in individual rehabilitation goals and center-specific practices and may have influenced outcomes. Future prospective studies should implement harmonized protocols to better characterize the dose–response relationship. Missing data further reduced the number of analyzable cases for some measures, although complete-case analysis was used without imputation. Fourth, only short-term outcomes were evaluated, and no long-term follow-up was available to assess whether improvements were sustained. Fifth, exclusion of patients with cognitive or communication impairments was based on therapists’ clinical judgment rather than a standardized screening tool, which may introduce variability between centers. Future studies could incorporate validated assessments to improve consistency and reproducibility. Finally, patient inclusion was restricted to subacute stroke, chosen because neuroplasticity and recovery potential are highest during this period [22], but this limited generalizability to acute and chronic populations [39].

Beyond methodological considerations, adoption bias should also be acknowledged. This study reflects early data collected from therapists motivated to trial a novel technology, at a time when no published clinical evidence on NeuroSkin^®^ was yet available. Early adoption often requires substantial clinician effort to learn new workflows, which may have influenced implementation. Moreover, as several authors are affiliated with the company developing NeuroSkin^®^, interpretation bias cannot be fully excluded, despite the involvement of seven independent rehabilitation centers. Future trials must include independent oversight and blinded analyses.

Despite these limitations, the results are encouraging. Improvements in gait outcomes exceeded clinically meaningful thresholds, usability was rated as excellent, and the system was safely deployed across multiple centers without disrupting routine workflows. This integration of AI-driven automation with therapist-guided clinical care demonstrates the potential for neurotechnology to scale safely and meaningfully into routine practice. These findings provide a foundation for larger, prospective, pre-registered randomized controlled trials to confirm efficacy, establish dose–response relationships, evaluate long-term effects, and systematically compare NeuroSkin^®^ with conventional FES approaches.

## 5. Conclusions

This multicenter retrospective feasibility study suggests that NeuroSkin^®^ can be implemented in real-world inpatient settings, with high therapist-reported usability and no reported adverse events. While improvements in gait-related outcomes were observed, these findings should be interpreted cautiously due to the uncontrolled design, small sample size, non-blinded assessments, and the absence of analyses adjusting for potential confounders. Larger prospective controlled studies are needed to confirm effectiveness, further characterize safety and tolerability, and evaluate integration into clinical workflows. These results nevertheless provide a foundation for designing prospective, multicenter randomized controlled trials, with harmonized protocols and long-term follow-up, to confirm efficacy and guide implementation strategies.

## Figures and Tables

**Figure 1 sensors-25-05614-f001:**
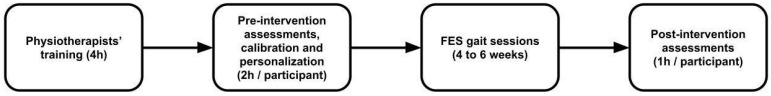
Clinical workflow of NeuroSkin^®^ integration in post-stroke rehabilitation. The process involves an initial training session, model personalization and intensity calibration, supervised FES-assisted gait therapy sessions, and post-intervention outcome assessment.

**Figure 2 sensors-25-05614-f002:**
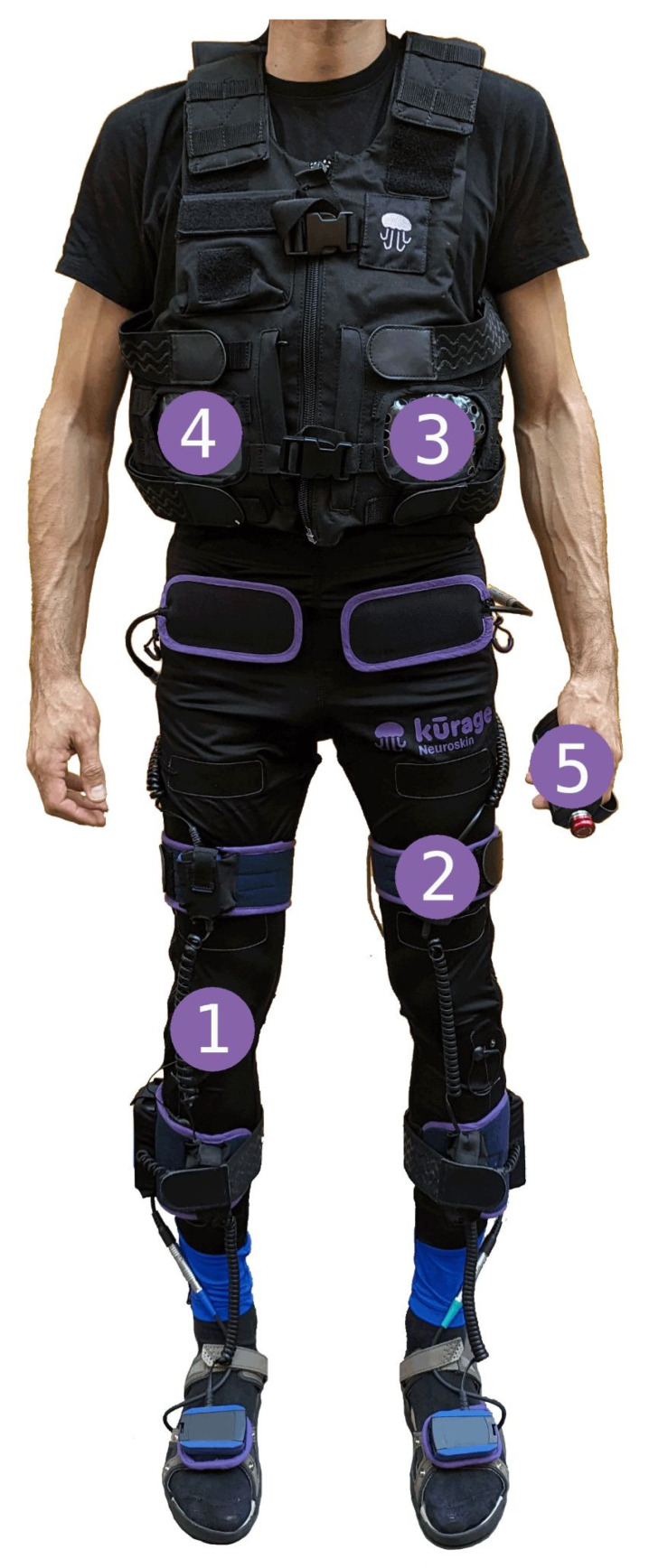
Components of the NeuroSkin^®^ system. The textile-based system includes embedded dry electrodes targeting key lower-limb muscle groups (1), inertial and force sensors for gait monitoring (2), an AI-driven gait phase detector (3), a programmable electrical stimulator (4), a remote controller used to regulate the overall intensity of stimulation (5), and a therapist-facing software for session management (see Figure 3).

**Figure 3 sensors-25-05614-f003:**
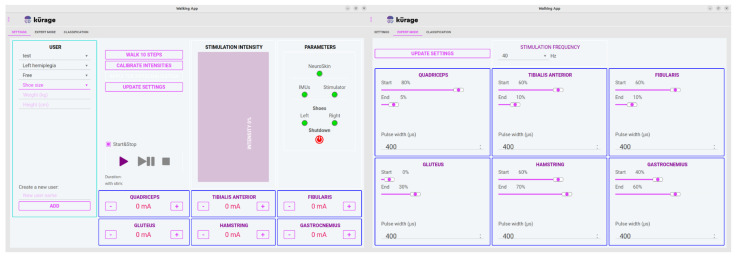
Graphical interface of the NeuroSkin^®^ application. The Global Settings screen (**left**) provides session control and patient profile management, while the Expert Settings screen (**right**) allows fine-tuning of stimulation parameters such as pulse width, intensity, and timing across muscle channels.

**Table 1 sensors-25-05614-t001:** Baseline demographic and clinical characteristics of the 15 post-stroke participants enrolled across the seven rehabilitation centers.

Patient	Age	Sex	Type of Stroke	Days Since Stroke	Paretic Side	Center	Number of Sessions
1	62	M	Ischemic	78	Right	1	20
2	71	F	Ischemic	89	Right	1	19
3	35	F	Hemorrhagic	60	Left	2	20
4	66	M	Ischemic	71	Left	3	11
5	49	F	Ischemic	65	Left	3	11
6	59	M	Ischemic	35	Right	4	15
7	72	M	Ischemic	29	Left	4	20
8	47	M	Ischemic	45	Right	5	20
9	72	M	Ischemic	85	Right	5	20
10	57	M	Hemorrhagic	29	Right	5	20
11	46	F	Ischemic	112	Left	5	20
12	72	M	Ischemic	42	Right	6	12
13	51	M	Hemorrhagic	86	Left	6	10
14	76	M	Ischemic	89	Left	7	12
15	74	F	Ischemic	10	Left	7	12

**Table 2 sensors-25-05614-t002:** Individual pre- and post-intervention results for the 10-Meter Walk Test (10MWT), 6-Minute Walk Test (6MWT), Timed Up and Go (TUG), and New Functional Ambulation Classification (NFAC), along with statistical analysis of clinical outcome measures before and after the intervention. Normality was assessed using the Shapiro–Wilk test; paired *t*-tests or Wilcoxon signed-rank tests were applied accordingly. The Mean Change row summarizes the evolution between baseline and post-intervention for each outcome (NFAC: mean absolute change in classification level; 10MWT and 6MWT: mean relative percentage improvement; TUG: mean relative percentage decrease in completion time). All measures showed statistically significant improvements with large effect sizes.

Patient	NFAC		10MWT		6MWT		TUG	
	Pre	Post	Pre	Post	Pre	Post	Pre	Post
1	3	7	0.56	0.83	197	303	-	-
2	2	6	0.2	0.45	63	270	-	-
3	2	5	0.21	0.43	71	143	54	32
4	3	6	0.57	1.05	155	375	30	13.45
5	2	5	0.16	0.24	43	67	45.53	39.07
6	3	7	1.01	1.21	322	367	8.99	7.26
7	2	5	0.45	0.72	23	230	47	16.58
8	2	5	-	-	28	200	170	15.4
9	2	6	0.44	1.02	135	250	27.4	13.63
10	4	7	0.64	0.66	210	210	21.26	18.06
11	5	6	1.03	1.8	356	394	11.28	9.69
12	6	7	0.7	1.04	245	495	11.26	8.56
13	1	2	0.1	0.17	45	50	191.28	48.58
14	5	6	0.29	0.56	85	125	24.89	23.41
15	1	8	0	0.97	0	351	unable	10.3
Normality	No		Yes		Yes		No	
(*p*-value)	(*p* = 0.0169)		(*p* = 0.0677)		(*p* = 0.1666)		(*p* = 0.0001301)	
Significance	Yes		Yes		Yes		Yes	
(*p*-value)	(*p* = 0.000632)		(*p* = 0.0004046)		(*p* = 0.0004718)		(*p* = 0.0004883)	
Effect size (Cohen’s d/r-value)	Large		Large		Large		Large	
	(r = 0.88)		(d = 1.26)		(d = 1.17)		(r = 1.01)	
**Mean change**	**+3**		**+70.3%**		**+176%**		**−39.1%**	

**Table 3 sensors-25-05614-t003:** Therapist-reported System Usability Scores (SUS) after the intervention.

**Operator**	1	2	3	4	5	6	7	**AVG**	STD
**SUS**	87.5	90	90	82.5	90	62.5	90	**84.6**	10.1

**Table 4 sensors-25-05614-t004:** Self-reported patient satisfaction after the intervention (1 = not satisfied at all; 10 = extremely satisfied).

**Patient**	1	2	3	4	5	6	7	8	9	10	11	12	13	14	15	**AVG**	STD
**SUS**	4	3	-	10	9	8	6	8	-	10	9	5	-	8	8	**7.33**	2.31

## Data Availability

All data have been provided in the article.

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
