# Peer review of "NeuroSkin®: AI-Driven Wearable Functional Electrical Stimulation for Post-Stroke Gait Recovery—A Multicenter Feasibility Study"

_sensors, 2025, doi:10.3390/s25185614_

Round 1
Reviewer 1 Report
Comments and Suggestions for Authors
This retrospective multicenter study evaluated the feasibility of an AI-driven wearable FES system (NeuroSkin) for gait rehabilitation in 15 subacute stroke patients in seven centers. Patients showed significant improvements in walking function after 10-20 sessions, and therapists rated the system as highly usable, while no adverse events were reported.
The manuscript is clearly written and easy to follow. However, I have several questions and suggestions that should be addressed prior to publication.
Major comments:
- The authors state that FES can enhance muscle recruitment, gait speed, and neuromuscular plasticity. However, the cited references are outdated and not specific to gait rehabilitation after stroke. I recommend including recent systematic reviews or meta-analyses that directly support the effectiveness of FES for post-stroke gait rehabilitation.
- Usability was assessed only by seven therapists. I consider this is a too small sample size to draw strong conclusions. Additionally, all therapists were trained and potentially influenced in the system capabilities, which may have introduced bias. Also, their level of experience and previous familiarity with technology use is not reported, which may also influence the reported usability outcomes.
- Usability from the patient's perspective was not assessed. In rehabilitation technology, both therapist and patient perspectives are important. The authors may consider applying or discussing additional tools such as the Quebec User Evaluation of Satisfaction with Assistive Technology (QUEST), Feasibility of Intervention Measure (FIM), or Acceptability of Intervention Measure (AIM). I understand that applying these scales may not be possible at this stage of the study, but you should mention this limitation in the discussion section.
- Although no adverse events were reported, the small sample size (n = 15), short intervention period, and retrospective design limit the ability to draw firm conclusions about safety. Moreover, patient-reported outcomes related to comfort or tolerability were not collected. Therefore, while the results are encouraging, they provide only an initial safety profile.
- The rationale for targeting subacute stroke patients, rather than acute or chronic patients, should be explicitly stated.
- One inclusion criterion was “responsiveness to FES.” How was this assessed prior to enrollment?
- One exclusion criterion was cognitive or communication impairment. Was a formal screening tool applied?
- The paragraph that is currently in the Results section (lines 201–212) describing outcome measures and statistical analysis should be moved to the Methods section.
It should also specify who performed the clinical assessments (10MWT, 6MWT, TUG, and NFAC), whether they were the same therapists that applied the FES protocol or someone else blinded to the study. Maybe also briefly describe what the scales measure and the rationale why they were selected as the outcome measures in your study.
- The meaning of “Evolution” in Table 2 should be clarified.
- Patient 15 in Table 12 has only one measure by outcome and this does not seem to be of the expected magnitude. This should be revised.
- The handling of missing data should be described as some patients have missing outcome measures.
- While the manuscript describes the number, duration, and general information about therapy sessions, it lacks sufficient detail for a clear understanding of the intervention. Specifically, the process for using the FES system in the patient and the structure of each therapy session. Information is missing such as the type of gait activities performed, session progression, rest periods, or real-time adjustments.
- Table 2 reports large effect sizes for changes in outcomes, but these are not discussed in depth. The authors should comment on the clinical relevance of these improvements and whether they exceed the minimal clinically important differences (MCIDs) for the clinical scales.
- The discussion should be strengthened by comparing the NeuroSkin system with other available FES devices in terms of feasibility, usability, safety, and effectiveness.
- Despite involving seven centers, the study enrolled only 15 patients. What were the difficulties you encountered in recruiting? This may also be valuable information to be discussed.
- Conclusions somewhat overstate the impact given the retrospective, uncontrolled nature of the study and small sample size. There is no analysis of possible confounders (e.g., time since stroke, number of sessions, baseline severity). The sample size may be too small to control for confounders or to make stratified analysis, but at least this should be discussed in the limitations.
- The limitations section should be strengthened by explicitly addressing the small sample sizes of both patients and therapists, as well as other methodological concerns noted in my previous commentaries.
Other minor suggestions:
Clarify the meaning of MR004 for readers unfamiliar with French data protection regulations.
In Figure 1, the final block should read “post-intervention” instead of “pre-intervention.”
Figure 2 may benefit from indicating the position of the components of the system in the image.
Figure 3 does not visualize complete.
In line 158, the sentence begins “NeuroSkin The data incoming...”. Either the word “NeuroSkin” is unnecessary or a verb is missing.
Author Response
Comment 1
The authors state that FES can enhance muscle recruitment, gait speed, and neuromuscular plasticity. However, the cited references are outdated and not specific to gait rehabilitation after stroke. I recommend including recent systematic reviews or meta-analyses that directly support the effectiveness of FES for post-stroke gait rehabilitation.
Response 1
We have updated the manuscript to include more recent and stroke-specific evidence supporting the efficacy of FES for gait rehabilitation. In particular, we now cite Chen et al. (2024), a systematic review and meta-analysis showing significant improvements in gait speed, step length, ankle dorsiflexion, and 6MWT outcomes when NMES was combined with rehabilitation programs. We also reference He et al. (2025), who demonstrated improved gait symmetry and joint kinematics using adaptive FES strategies. Lastly, we cite Howlett's et al. (2015) meta-analysis confirming modest but significant gains in walking speed and functional activity levels. These additions have been included in the introduction through the following paragraph to provide a stronger and up-to-date evidence base for the effectiveness of FES in post-stroke gait recovery.:
Recent systematic reviews and meta-analyses strengthen this evidence in post-stroke populations, with reported significant improvements in gait speed, cadence, step length, ankle dorsiflexion, and 6MWT distance when NMES was integrated into rehabilitation programs; enhanced biomechanical gait parameters, including COP symmetry and joint kinematics; and significant improvements in walking speed and functional activity levels compared with conventional rehabilitation. Together, these findings provide strong support for the use of FES in post-stroke gait recovery.
Comment 2
Usability was assessed only by seven therapists. I consider this is a too small sample size to draw strong conclusions. Additionally, all therapists were trained and potentially influenced in the system capabilities, which may have introduced bias. Also, their level of experience and previous familiarity with technology use is not reported, which may also influence the reported usability outcomes.
Response 2
We agree that the small number of therapists limits the generalizability of the usability findings. To address this, we have clarified in the Results section that the SUS score reflects high perceived usability among trained therapists, rather than strong evidence of universal usability. The following sentence:
The mean SUS score was 84.6, indicating excellent usability
has been replaced by:
The mean SUS score was 84.6, indicating high perceived usability among trained therapists within this feasibility context
Furthermore, we have updated the Discussion to explicitly acknowledge two potential sources of bias:
The following sentence:
Limitations include the retrospective nature of the study, lack of a control group, and inter-site protocol variability.
has been expanded to:
Limitations include the retrospective nature of the study, lack of a control group, and inter-site protocol variability. In addition, specific limitations should be noted when interpreting the usability findings. First, usability was assessed by only seven therapists, consistent with the small-sample design typical of feasibility studies, but limiting generalizability. Second, all therapists underwent NeuroSkin®-specific training prior to clinical use. While necessary to ensure correct device handling, this may have introduced a positive bias in their ratings. Finally, although none of the therapists had prior experience with FES, information regarding their general clinical experience and familiarity with digital technologies was not collected, which further constrains interpretation of the usability outcomes.
We expect that these additions will provide better context for interpreting the usability results and highlight the need for further validation in larger, more heterogeneous samples.
Comment 3
Usability from the patient's perspective was not assessed. In rehabilitation technology, both therapist and patient perspectives are important. The authors may consider applying or discussing additional tools such as the Quebec User Evaluation of Satisfaction with Assistive Technology (QUEST), Feasibility of Intervention Measure (FIM), or Acceptability of Intervention Measure (AIM). I understand that applying these scales may not be possible at this stage of the study, but you should mention this limitation in the discussion section.
Response 3
We agree that incorporating patient perspectives is critical for evaluating rehabilitation technologies comprehensively. In this feasibility study, usability was assessed exclusively from the therapists’ perspective to focus on workflow integration during early-stage deployment. While we did not apply standardized usability questionnaires with the patients, we did actually collect a preliminary measure of patient satisfaction using a single-item numeric scale from 1 (not satisfied) to 10 (extremely satisfied). Among the 12 patients who responded (3 did not answer), the mean satisfaction score was 7.33 ± 2.31, suggesting a generally positive perception of the intervention. We initially didn’t include this in the manuscript because of the following major limitations:
- the use of a simple non-validated measure
- potential social desirability bias
- dependence on the therapist or center administering the intervention
However in the light of your comment, we have decided to mention it, because even though the method was simplistic, it shows we considered patient satisfaction. We have added a table (n°4) to the Results section after the SUS results along with the following paragraph:
In addition, patient satisfaction with the intervention was assessed at the end of the program using a single-item numeric rating scale (1 = not satisfied at all; 10 = extremely satisfied). Among the 12 patients who responded (3 non-respondents), the mean satisfaction score was 7.33 ± 2.31. While these preliminary findings suggest a generally positive perception of the therapy, the simplicity of the measure and potential response biases limit their interpretability.
and we clarified in the Discussion that future studies should incorporate validated tools such as QUEST, FIM, or AIM to more comprehensively capture patient-reported usability and acceptability, by adding the following:
Although patient satisfaction was collected, it was assessed using a simple single-item numeric scale, which provides only a rough estimate and does not capture specific dimensions of usability or acceptability. Moreover, responses may have been influenced by social desirability bias and by the local therapists administering the intervention. Future studies should incorporate validated patient-reported measures such as QUEST, FIM, or AIM to better evaluate patient experience.
Comment 4
Although no adverse events were reported, the small sample size (n = 15), short intervention period, and retrospective design limit the ability to draw firm conclusions about safety. Moreover, patient-reported outcomes related to comfort or tolerability were not collected. Therefore, while the results are encouraging, they provide only an initial safety profile.
Response 4
We agree that safety conclusions should remain preliminary. We have updated the manuscript accordingly. In the end of the Results section, we now specify that the absence of reported adverse events provides only an initial safety profile:
No adverse events or device-related complications were reported during the intervention. However, given the small sample size and short intervention period, these findings provide only an initial safety profile.
Additionally, we have added the following in the Discussion:
While no adverse events were reported, the retrospective design, small sample size, and short duration of intervention limit the ability to draw definitive conclusions about safety. In addition, patient-reported outcomes related to comfort or tolerability were not collected, and future studies should address this gap to better characterize the user experience.
Comment 5
The rationale for targeting subacute stroke patients, rather than acute or chronic patients, should be explicitly stated.
Response 5
Subacute patients were selected for multiple reasons:
- Neuroplasticity and recovery potential are highest during the subacute phase, making it an optimal window to evaluate gait rehabilitation technologies.
- Clinical priorities: gait training is typically emphasized during this stage as part of intensive inpatient rehabilitation programs.
- Feasibility constraints: acute patients are often unable to walk safely at this early stage and many remain medically unstable, while chronic patients’ functional recovery often plateaus and findings would be less generalizable to standard inpatient care.
- Pragmatic considerations: the participating rehabilitation centers primarily admitted subacute inpatients; focusing on this population helped reduce variability and improve interpretability of findings given the limited sample size.
Thus, the following paragraph was inserted after the inclusion criteria:
Subacute stroke patients were specifically targeted for this feasibility study, which was motivated by the following factors. Acute patients were excluded because few are able to walk safely at this stage and many remain medically unstable, limiting the feasibility of FES-assisted gait training. From a neurophysiological standpoint, spontaneous neuroplasticity and functional recovery potential are highest during the acute and subacute phase, making it an optimal period for evaluating rehabilitation interventions. Clinically, gait training is typically prioritized during this stage, as most rehabilitation programs intensify mobility-focused therapy during early recovery. Chronic patients were also excluded, as functional recovery generally plateaus beyond this period. Finally, from a pragmatic standpoint, the participating rehabilitation centers primarily admitted subacute inpatients; focusing on this population reduced variability and improved interpretability of findings given the limited sample size.
And the following comment has been added in the discussion:
Because the study focused exclusively on subacute patients, the results may not generalize directly to acute or chronic stroke populations, which should be explored in future studies.
Comment 6
One inclusion criterion was “responsiveness to FES.” How was this assessed prior to enrollment?
Response 6
Responsiveness to FES was assessed before enrollment by applying a brief stimulation using a surface TENS stimulator on the quadriceps muscle. Patients were considered responsive if the stimulation elicited a visible contraction strong enough to extend the leg against gravity. This has now been specified in the Materials and Methods / Study Design section, after explaining why subacute patients have been targeted.
Comment 7
One exclusion criterion was cognitive or communication impairment. Was a formal screening tool applied?
Response 7
No formal cognitive or communication screening tool was applied; instead, eligibility was determined based on the treating therapists’ clinical judgment regarding the patients’ ability to understand instructions and complete assessments. We have clarified this in the Materials and Methods section, and we have also acknowledged in the Discussion that this approach may introduce variability, and that future studies should consider incorporating standardized tools for improved consistency.
Response 8
The paragraph that is currently in the Results section (lines 201–212) describing outcome measures and statistical analysis should be moved to the Methods section. It should also specify who performed the clinical assessments (10MWT, 6MWT, TUG, and NFAC), whether they were the same therapists that applied the FES protocol or someone else blinded to the study. Maybe also briefly describe what the scales measure and the rationale why they were selected as the outcome measures in your study.
Response 8
We have moved the description of outcome measures and statistical analysis from the Results section to the Materials and Methods section under a dedicated subsection titled ‘Outcome Measures and Statistical Analysis’ to improve the clarity and logical structure of the manuscript. This section:
- Describes what each of the four outcome measures assesses
- Clarifies that these measures were not specifically selected for this study but were part of the standard clinical routine at the participating centers and were therefore used for the retrospective analysis, which also facilitates comparison with existing literature.
- Clarifies that all assessments were performed by the same physiotherapists who administered the FES sessions and that no blinded assessors were involved
We have also acknowledged in the Discussion that the absence of blinded assessors is a limitation to be addressed in future prospective trials.
Comment 9
The meaning of “Evolution” in Table 2 should be clarified.
Response 9
We have renamed it to “Mean Change” and inserted the following sentence in the table caption:
The Mean Change row summarizes the evolution between baseline and post-intervention for each outcome: NFAC: mean absolute change in classification level; 10MWT and 6MWT: mean relative percentage improvement; TUG: mean relative percentage decrease in completion time.
Comment 10
Patient 15 in Table 12 has only one measure by outcome and this does not seem to be of the expected magnitude. This should be revised.
Response 10
This typo was corrected
Comment 11
The handling of missing data should be described as some patients have missing outcome measures
Response 11
We have clarified in the Materials and Methods that missing data were handled using a complete-case analysis approach: for each outcome measure, only patients with both baseline and post-intervention values were included in the corresponding statistical tests, and no data imputation was performed. We have also acknowledged in the Discussion that missing data may limit the robustness of some analyses and should be minimized in future prospective studies.
Comment 12
While the manuscript describes the number, duration, and general information about therapy sessions, it lacks sufficient detail for a clear understanding of the intervention. Specifically, the process for using the FES system in the patient and the structure of each therapy session. Information is missing such as the type of gait activities performed, session progression, rest periods, or real-time adjustments.
Response 12
We have added a section titled “Intervention” in the Materials and Methods section to provide additional details on how the NeuroSkin® system was used during therapy sessions. It includes:
- The personalization procedure and calibration of stimulation intensities
- Typical gait training activities performed during sessions
- The general session structure, including walking bouts, rest periods, and individualized progression
- The possibility for real-time adjustments of stimulation parameters during therapy
We also emphasized that session protocols were intentionally individualized to reflect real-world clinical workflows, which inevitably results in variability between centers that have been acknowledged as a limitation in the Discussion.
Comment 13
Table 2 reports large effect sizes for changes in outcomes, but these are not discussed in depth. The authors should comment on the clinical relevance of these improvements and whether they exceed the minimal clinically important differences (MCIDs) for the clinical scales.
Response 13
We have expanded the Results and Discussion sections to interpret the observed improvements in the context of clinically meaningful thresholds. The main findings are as follows:
- 10MWT: mean increase of +0.41 m/s, exceeding both the small (0.06 m/s) and substantial (0.14 m/s) MCIDs reported by Perera et al. (2006).
- 6MWT: median improvement of ~210 m (+176%), far above the MCID of 18.6 m or 4.8% defined by Flansbjer et al. (2005).
- TUG: mean reduction of ~15 s (−39%), surpassing both the minimal detectable change (2.9 s) and smallest real difference threshold (23%) reported by Flansbjer et al. (2005).
- NFAC: no validated MCID exists, but the mean gain of three categories indicates substantial functional improvement.
These additions emphasize that the observed changes are likely to be clinically meaningful, while also acknowledging that thresholds vary across studies and that confirmation in larger, prospective trials is needed.
Added in the Results:
The observed improvements exceeded published thresholds for clinically meaningful change. The mean increase in gait speed (+0.35 m/s) surpassed both the small (0.06 m/s) and substantial (0.14 m/s) MCIDs as reported in [19]. Similarly, the median improvement in 6MWT distance (+123,5 m) was far greater than the MCID of 18.6 m identified in [20]. For TUG, although no MCID is available, the observed mean reduction (36,9 s) exceeded both the minimal detectable change (2.9 s) and the smallest real difference threshold (23%) reported in [20]. No validated MCID exists for the NFAC scale, but the mean gain of three categories strongly suggests meaningful improvements in walking autonomy.
Added in the discussion:
Beyond statistical significance, the magnitude of observed changes suggests a clinically meaningful impact. Improvements in gait speed, walking endurance, and mobility substantially exceeded published MCIDs or MDC/SRD thresholds for post-stroke populations. While the NFAC scale lacks an established MCID, the average increase of three categories likely reflects relevant functional gains. These findings support the potential of AI-driven, FES-assisted gait training to produce changes that are meaningful for patients’ everyday activities.
Comment 14
The discussion should be strengthened by comparing the NeuroSkin system with other available FES devices in terms of feasibility, usability, safety, and effectiveness.
Response 14
We would like to emphasize that NeuroSkin® differs fundamentally from existing peroneal-nerve FES devices: it provides multi-channel, phase-specific stimulation across six muscle groups, whereas most established systems deliver single- or dual-channel stimulation to address isolated foot drop. Furthermore, most published studies on FES for post-stroke gait rehabilitation evaluate single-channel peroneal-nerve devices designed to correct foot drop. This makes direct comparisons regarding feasibility and usability less straightforward. Regarding safety, the underlying stimulation principles are the same, and no device-specific risks are expected. Finally, since this study was a retrospective feasibility analysis without a control group, effectiveness comparisons with other devices would be beyond its scope.
Nonetheless, we have added the following clarifying statement in the Discussion in order to explicitly note these distinctions and highlight that comparative efficacy and usability should be explored in future prospective trials:
Direct comparisons between NeuroSkin® and existing peroneal-nerve FES devices (e.g., WalkAide, L300, ODFS) should be interpreted cautiously. Most available studies on FES for gait rehabilitation focus on single or dual-channel devices designed to stimulate the common peroneal nerve and correct isolated foot drop. In contrast, NeuroSkin® delivers synchronized, phase-specific stimulation to six lower-limb muscle groups using AI-based gait-phase detection, enabling broader neuromuscular engagement but inherently increasing system complexity. While this difference may affect feasibility and usability, safety considerations are similar across FES modalities, as low-level electrical stimulation is well established. Finally, because this is a retrospective feasibility study without a control group, effectiveness comparisons with other devices would be premature. Future prospective trials should address comparative efficacy, usability, and workflow integration.
Comment 15
Despite involving seven centers, the study enrolled only 15 patients. What were the difficulties you encountered in recruiting? This may also be valuable information to be discussed.
Response 15
We did not encounter recruitment difficulties; the limited sample size reflects the early stage of product deployment rather than patient availability. At the time of data collection (2023), NeuroSkin® was a novel system without published evidence directly evaluating its use, and early adoption was limited to a small group of trained therapists willing to integrate a new device into their workflows. We have clarified this in the end of the Discussion and noted that these findings represent preliminary data from early adopters, ahead of larger prospective and controlled studies currently planned:
Although this study involved seven centers, only 15 patients were included. This reflects the early stage of product deployment rather than recruitment difficulties. Because NeuroSkin® was a novel system at the time, adoption was limited to a small group of trained therapists motivated to experiment with new technologies. Early adoption of innovative rehabilitation devices often requires significant effort from clinicians to learn new workflows, particularly when device-specific evidence is not yet published. The present analysis therefore reports data collected from these early adopters, with larger prospective and controlled studies planned to further evaluate effectiveness and broader clinical integration
Comment 16
Conclusions somewhat overstate the impact given the retrospective, uncontrolled nature of the study and small sample size. There is no analysis of possible confounders (e.g., time since stroke, number of sessions, baseline severity). The sample size may be too small to control for confounders or to make stratified analysis, but at least this should be discussed in the limitations.
Response 16
We have revised the Conclusion to emphasize feasibility and to avoid overstating impact:
This multicenter retrospective feasibility study suggests that NeuroSkin® can be implemented in real-world inpatient settings, with high therapist-reported usability and no reported adverse events. While improvements in gait-related outcomes were observed, these findings should be interpreted cautiously due to the uncontrolled design, small sample size, non-blinded assessments, and the absence of analyses adjusting for potential confounders (e.g., time since stroke, number of sessions, baseline severity). Larger prospective controlled studies are needed to confirm effectiveness, further characterize safety and tolerability, and evaluate integration into clinical workflows.
Comment 17
The limitations section should be strengthened by explicitly addressing the small sample sizes of both patients and therapists, as well as other methodological concerns noted in my previous commentaries.
Response 17
Thanks to the previous comments, we have already addressed the following methodological concerns:
- Small number of therapists was acknowledged, with implications for generalizability.
- Bias in usability ratings has been covered.
- Missing patient-reported usability measures were addressed.
- Safety profile preliminary was acknowledged.
- No blinded assessors were stated.
- Missing outcome measures were mentioned.
- Subacute-only population was addressed.
In addition, we have strengthened the Limitations section to explicitly state that the small sample sizes of both patients and therapists reduce generalizability and preclude analyses controlling for potential confounders such as time since stroke, number of sessions, and baseline severity. To do so, we added the following opening paragraph to the limitation section:
The main limitation of this study is the small sample size of both patients and therapists, which reduces statistical power, limits generalizability, and precludes multivariable analyses to control for potential confounding factors such as time since stroke, baseline severity, and number of sessions.
Other minor suggestions
Clarify the meaning of MR004 for readers unfamiliar with French data protection regulations.
The following statement has been added to the Study Design section: Ethics compliance and data collection and analysis were carried out in accordance with the MR004 reference methodology, which governs the processing of personal health data for non-interventional retrospective research. Under MR004, such studies require declaration to the French Data Protection Authority (CNIL). In addition, all patients or legal guardians provided informed consent for treatment and use of anonymized data.
In Figure 1, the final block should read “post-intervention” instead of “pre-intervention.”
Corrected
Figure 2 may benefit from indicating the position of the components of the system in the image.
Added
Figure 3 does not visualize complete.
Corrected
In line 158, the sentence begins “NeuroSkin The data incoming...”. Either the word “NeuroSkin” is unnecessary or a verb is missing.
Corrected
Reviewer 2 Report
Comments and Suggestions for Authors
Congratulations on developing a study on this interesting device. It is innovative and clinically useful for the treatment of gait disturbances after stroke. It is described in detail in the article. However, the sample size is small and the study design is limited: an experimental or quasi-experimental design would have provided more robust results. To improve the manuscript, I suggest expanding the "Introduction" and "Discussion" sections by including previous studies in which the same variables were measured and conventional FES was used. This would help more clearly highlight the novelty of the device.
Below, I leave my suggestions to improve the manuscript:
Introduction
- I recommend providing reference to specific studies that support these statements (lines 56-58).
- If available in the literature, it would be appropriate to include references introducing and supporting NeuroSkin.
- I suggest expanding the introduction by mentioning previous studies where conventional FES is used for gait reeducation.
Methods
- There is considerable heterogeneity in the number of sessions carried out, which inevitably affects the findings regarding the intervention’s effectiveness.
- Since Neuroskin training is integrated into conventional physiotherapy treatment, it would be helpful to specify the duration of the conventional treatment for each patient and whether these were similar across participants, as this could be a potential source of bias in the results.
- The duration of training course for physiotherapists is 3.5 or 4 hours? This information is inconsistent between Figure 1 and the main text.
- Line 104: average, no median OR median and IQR.
- A brief description of the variables measured is missing (lines 201–203 could be moved here and the outcomes should be explained). Additionally, the SUS questionnaire should be described in more detail, including how the scores are interpreted, as it represents the most objective measure of feasibility in this study.
- It is necessary to move the explanation of the statistical analysis (203-212) to this section.
Results
- Line 196: It is better to report the percentage of female and male, not fractions.
- A more detailed paragraph presenting the most relevant results from the statistical analysis is needed, as currently these are only presented in the table.
Discussion
This section should be further developed, comparing the findings with previous studies that evaluated the same variables and used conventional FES systems. The results related to the SUS scale should also be further explored and compared with the findings of similar studies, as this is a key measure of feasibility in the present work.
Author Response
INTRODUCTION
1
I recommend providing reference to specific studies that support these statements (lines 56-58).
- Several systematic reviews and clinical reports highlight that most commercially available FES devices are either overly simplistic (e.g., single-channel foot-drop correction) or technically demanding, requiring manual electrode placement, patient-specific calibration, and therapist training [Prenton 2016; Howlett 2015; Kottink 2012; NICE 2017]. These workflow complexities and usability barriers have been reported as key reasons why FES remains underused in routine stroke rehabilitation despite demonstrated efficacy.
- We have added these references and updated the statement as follows: Most commercially available FES systems are either overly simplistic (e.g., single-channel devices addressing isolated foot drop) or highly technical, requiring manual electrode placement, individual calibration, and therapist training. These workflow complexities and steep learning curves have contributed to FES being underused in routine post-stroke rehabilitation.
2
If available in the literature, it would be appropriate to include references introducing and supporting NeuroSkin
- We have cited Feppon et al. (2025), which provides the technical characterization of NeuroSkin as a textile-based, multi-channel FES system with embedded movement sensors and dry electrodes. As published literature on NeuroSkin® is still limited, this feasibility study represents one of the first multicenter clinical evaluations of its integration into routine rehabilitation practice. This clarification has been added to both the Introduction and Methods sections
3
I suggest expanding the introduction by mentioning previous studies where conventional FES is used for gait reeducation
- We have updated the manuscript to include more recent and stroke-specific evidence supporting the efficacy of FES for gait rehabilitation. In particular, we now cite Chen et al. (2024), a systematic review and meta-analysis showing significant improvements in gait speed, step length, ankle dorsiflexion, and 6MWT outcomes when NMES was combined with rehabilitation programs. We also reference He et al. (2025), who demonstrated improved gait symmetry and joint kinematics using adaptive FES strategies. Lastly, we cite Howlett's et al. (2015) meta-analysis confirming modest but significant gains in walking speed and functional activity levels. The following paragraph has been inserted in the Introduction to provide a stronger and up-to-date evidence base for the effectiveness of FES in post-stroke gait recovery.:
Recent systematic reviews and meta-analyses strengthen this evidence in post-stroke populations, with reported significant improvements in gait speed, cadence, step length, ankle dorsiflexion, and 6MWT distance when NMES was integrated into rehabilitation programs; enhanced biomechanical gait parameters, including COP symmetry and joint kinematics; and significant improvements in walking speed and functional activity levels compared with conventional rehabilitation. Together, these findings provide strong support for the use of FES in post-stroke gait recovery.
METHODS
4
There is considerable heterogeneity in the number of sessions carried out, which inevitably affects the findings regarding the intervention’s effectiveness.
- We have clarified in the Methods that the number of FES-assisted sessions varied (10–20) according to local clinical protocols, patient-specific rehabilitation goals, and length of stay. We have also acknowledged in the Limitations that this heterogeneity may have influenced the findings and that the sample size was insufficient to analyze dose-response effects. Future prospective studies will use harmonized intervention protocols to better characterize optimal session dosage.
5
Since Neuroskin training is integrated into conventional physiotherapy treatment, it would be helpful to specify the duration of the conventional treatment for each patient and whether these were similar across participants, as this could be a potential source of bias in the results.
- We have added a section (2.2. Intervention) in the Materials and Methods section to provide additional details on how the NeuroSkin® system was used during therapy sessions. It includes:
- The personalization procedure and calibration of stimulation intensities
- Typical gait training activities performed during sessions
- The general session structure, including walking bouts, rest periods, and individualized progression
- The possibility for real-time adjustments of stimulation parameters during therapy
We also emphasized that session protocols were intentionally individualized to reflect real-world clinical workflows, which may result in some variability between centers that have been acknowledged as a limitation in the Discussion. - All participants received conventional physiotherapy in addition to the FES-assisted gait training with NeuroSkin®. On average, patients followed 1 to 2 hours of standard physiotherapy per day, five days a week, during their inpatient rehabilitation stay. This included gait training, balance exercises, and functional mobility tasks.
- Although session content was adapted to each patient’s needs—as is typical in real-world rehabilitation—the total daily therapy duration was relatively consistent across centers due to institutional routines. We acknowledge, however, that variations in treatment content and intensity could represent a source of inter-subject variability, and we have now addressed this as a potential limitation in the Discussion section.
- The following sentence was added to the Study Design section: In addition to the NeuroSkin® intervention, all patients received 1 to 2 hours of standard physiotherapy per day, five days a week, in accordance with usual care protocols at each center.
- And the following was added to the discussion: Although daily physiotherapy duration was relatively consistent across participants, variation in the content and intensity of conventional rehabilitation may have contributed to differences in patient outcomes and represents a potential source of bias.
6
The duration of training course for physiotherapists is 3.5 or 4 hours? This information is inconsistent between Figure 1 and the main text.
- This has been corrected to 4h
7
Line 104: average, no median OR median and IQR.
- We have replaced median by average
8
A brief description of the variables measured is missing (lines 201–203 could be moved here and the outcomes should be explained).
- We have moved the description of outcome measures and statistical analysis from the Results section to the Materials and Methods section under a dedicated 2.3 subsection titled ‘Outcome Measures and Statistical Analysis’ to improve the clarity and logical structure of the manuscript. This sectio:
- Describes what each of the four outcome measures assesses
- Clarifies that these measures were not specifically selected for this study but were part of the standard clinical routine at the participating centers and were therefore used for the retrospective analysis, which also facilitates comparison with existing literature.
- Clarifies that all assessments were performed by the same physiotherapists who administered the FES sessions and that no blinded assessors were involved
- We have also acknowledged in the Discussion that the absence of blinded assessors is a limitation to be addressed in future prospective trials.
9
Additionally, the SUS questionnaire should be described in more detail, including how the scores are interpreted, as it represents the most objective measure of feasibility in this study.
- We have added a « Usability and Satisfaction » section in the Methods to include details on SUS structure, scoring, and interpretation thresholds. We also added that, in addition to the SUS, patients completed a basic satisfaction survey providing a single numeric rating from 1 to 10 of their overall experience.
10
It is necessary to move the explanation of the statistical analysis (203-212) to this section.
- Done
RESULTS
11
Line 196: It is better to report the percentage of female and male, not fractions.
- To avoid writing 33.33% and 66.66%, we have instead simply written 5 female and 10 male
12
A more detailed paragraph presenting the most relevant results from the statistical analysis is needed, as currently these are only presented in the table.
- We have expanded the Results and Discussion sections to interpret the observed improvements in the context of clinically meaningful thresholds. The main findings are as follows:
- 10MWT: mean increase of +0.41 m/s, exceeding both the small (0.06 m/s) and substantial (0.14 m/s) MCIDs reported by Perera et al. (2006).
- 6MWT: median improvement of ~210 m (+176%), far above the MCID of 18.6 m or 4.8% defined by Flansbjer et al. (2005).
- TUG: mean reduction of ~15 s (−39%), surpassing both the minimal detectable change (2.9 s) and smallest real difference threshold (23%) reported by Flansbjer et al. (2005).
- NFAC: no validated MCID exists, but the mean gain of three categories indicates substantial functional improvement. These additions emphasize that the observed changes are likely to be clinically meaningful, while also acknowledging that thresholds vary across studies and that confirmation in larger, prospective trials is needed. - We added the following paragraph in the Results section:
The observed improvements exceeded published thresholds for clinically meaningful change. The mean increase in gait speed (+0.35 m/s) surpassed both the small (0.06 m/s) and substantial (0.14 m/s) MCIDs as reported in [19]. Similarly, the median improvement in 6MWT distance (+123,5 m) was far greater than the MCID of 18.6 m identified in [20]. For TUG, although no MCID is available, the observed mean reduction (36,9 s) exceeded both the minimal detectable change (2.9 s) and the smallest real difference threshold (23%) reported in [20]. No validated MCID exists for the NFAC scale, but the mean gain of three categories strongly suggests meaningful improvements in walking autonomy.
Added in the discussion:
Beyond statistical significance, the magnitude of observed changes suggests a clinically meaningful impact. Improvements in gait speed, walking endurance, and mobility substantially exceeded published MCIDs or MDC/SRD thresholds for post-stroke populations. While the NFAC scale lacks an established MCID, the average increase of three categories likely reflects relevant functional gains. These findings support the potential of AI-driven, FES-assisted gait training to produce changes that are meaningful for patients’ everyday activities.
DISCUSSION
13
This section should be further developed, comparing the findings with previous studies that evaluated the same variables and used conventional FES systems. The results related to the SUS scale should also be further explored and compared with the findings of similar studies, as this is a key measure of feasibility in the present work.
- We have expanded the Discussion to
- compare our findings with previous studies using conventional FES systems. While gait improvements observed in this study are broadly consistent with prior reports on peroneal-nerve FES [Prenton 2016; Howlett 2015], direct comparisons are limited given that most existing studies focus on single-channel foot-drop correction, whereas NeuroSkin® delivers multi-channel, phase-specific stimulation.
- compare our SUS findings with those from an existing usability study of the Fesia Walk device. In that study, therapists rated usability as “excellent,” with a mean SUS score of 85.6 /100. Our mean SUS score of 84.6—achieved by NeuroSkin®, which delivers multi-channel, AI-driven phase-specific stimulation—suggests that our system delivers usability on par with existing multi-field systems.
Reviewer 3 Report
Comments and Suggestions for Authors
The NeuroSkin® system, an AI-driven wearable Functional Electrical Stimulation (FES) device for post-stroke gait rehabilitation, has been the subject of a multicenter feasibility study. The system is innovative and timely, addressing the limitations of current FES systems such as complexity and limited clinical scalability. The system received excellent usability scores and no adverse events were reported. Data collection from seven rehabilitation centers enhanced generalizability. However, there are several areas that warrant improvement or clarification to enhance scientific rigor and transparency.
The study's strengths include its innovative integration of AI-driven gait phase detection and wearable dry electrodes, excellent usability and safety scores, and comprehensive system description. However, the absence of a control group limits the ability to attribute observed improvements directly to NeuroSkin®. Future studies should include randomized controlled designs to establish causal inference and assess relative efficacy.
The retrospective design limits control over confounders and introduces potential biases. A prospective, pre-registered protocol would improve methodological robustness. Inconsistencies in data reporting, such as missing TUG results for some patients, need to be corrected. Conflict of interest between multiple authors employed by the company that manufactures NeuroSkin® may influence interpretation.
The study only evaluates short-term outcomes, so it should assess whether improvements are sustained over time. Minor criticisms include statistical reporting, terminology and abbreviations, and AI model validation.
In conclusion, the NeuroSkin® system is a promising tool for post-stroke gait rehabilitation, but its scientific validity is limited by methodological weaknesses. It should be followed by a rigorously designed prospective randomized controlled trial.
Author Response
1
The absence of a control group limits the ability to attribute observed improvements directly to NeuroSkin®. Future studies should include randomized controlled designs to establish causal inference and assess relative efficacy.
- We agree with the reviewer and have clarified this point in several sections of the manuscript. In both the Abstract and Discussion, we now emphasize that this study is a retrospective feasibility analysis and that the observed improvements cannot be causally attributed to NeuroSkin®. We also highlight in the Limitations that the absence of a control group prevents establishing causal inference and that prospective randomized controlled trials are planned to assess relative efficacy.
2
The retrospective design limits control over confounders and introduces potential biases. A prospective, pre-registered protocol would improve methodological robustness.
- We agree with the reviewer and have addressed this point in the Discussion and Limitations. We now explicitly state that the retrospective design limited control over potential confounders (e.g., time since stroke, baseline severity, and number of sessions) and introduced possible biases. We also note that a prospective, pre-registered randomized controlled trial is planned to improve methodological robustness and confirm these preliminary findings.
3
Inconsistencies in data reporting, such as missing TUG results for some patients, need to be corrected.
- We have clarified in the Materials and Methods that a complete-case analysis approach was used: only patients with both baseline and post-intervention values for a given outcome were included, and no data imputation was performed. We also note this in the Limitations, acknowledging that missing data reduced the number of available data points for certain measures.
- Patient 15 in Table 12 had only one measure by outcome, this typo was corrected.
4
Conflict of interest between multiple authors employed by the company that manufactures NeuroSkin® may influence interpretation.
- Indeed, this is an important point. We have clarified this potential conflict in two ways:
- We have refined the conflict of interest statement
- We have stated in the discussion that because of this conflict of interest, potential bias cannot be entirely excluded; however the study involved seven independent rehabilitation centers where data were collected locally, and future studies will incorporate independent oversight and blinded analyses to further minimize potential bias.
5
The study only evaluates short-term outcomes, so it should assess whether improvements are sustained over time.
- We agree with the reviewer and have clarified this in the Discussion: Because this was a retrospective feasibility study, only short-term outcomes were available, and we cannot determine whether the observed improvements were sustained. We have noted in the Limitations that future prospective controlled studies should include long-term follow-up to evaluate the durability of these effects.
6
Minor criticisms include statistical reporting, terminology and abbreviations, and AI model validation.
- We have carefully reviewed the manuscript to ensure:
- Statistical reporting is consistent and complete, including explicit naming of statistical tests, reporting of p-values, and effect sizes.
- Terminology and abbreviations are clearly defined at first mention and used consistently throughout.
- AI model validation is now clarified in the Methods’ dedicated section “AI Model and Personalization procedure” where we specify that NeuroSkin uses a pre-trained gait-phase detection model refined during personalization, and we cite Feppon et al. (2025) for technical validation details.
7
The NeuroSkin® system is a promising tool for post-stroke gait rehabilitation, but its scientific validity is limited by methodological weaknesses. It should be followed by a rigorously designed prospective randomized controlled trial.
- We have clarified throughout the manuscript that this study represents a retrospective feasibility analysis and that the findings should be considered preliminary. In both the Discussion and Conclusions, we now explicitly state that a rigorously designed prospective randomized controlled trial is planned to confirm efficacy, control for potential confounders, and assess long-term outcomes.
Reviewer 4 Report
Comments and Suggestions for Authors
This paper resembles a performance report on NeuroSkin®, a wearable FES system that uses AI-driven stimulation and sensor-based gait monitoring. It lacks clear structure, motivation, and contribution. Therefore, it does not meet the scientific standards of Sensors Journal and should be rejected.
Author Response
This paper resembles a performance report on NeuroSkin®, a wearable FES system that uses AI-driven stimulation and sensor-based gait monitoring. It lacks clear structure, motivation, and contribution. Therefore, it does not meet the scientific standards of Sensors Journal and should be rejected.
We thank the reviewer for their assessment and would like to clarify several points:
- Scientific Contribution: This is the first multicenter clinical feasibility study of an AI-driven, multi-channel, textile-based FES system for gait rehabilitation after stroke. To our knowledge, no prior studies have evaluated such a system in a real-world clinical setting across several independent rehabilitation centers. We now explicitly state this at the end of the Introduction
- Objective Evaluation: The study includes standardized, validated clinical outcome measures (10MWT, 6MWT, TUG, NFAC) and the System Usability Scale (SUS), providing quantitative and interpretable results rather than a device performance report.
- Motivation & Structure: Following the suggestions of other reviewers, we have strengthened the Introduction, clarified study objectives, expanded the Discussion to position our findings within the existing FES literature, and refined the Limitations to transparently present methodological constraints and future directions.
We believe these revisions have substantially improved the clarity, scientific rigor, and contextualization of the work, hence we submit the revised manuscript for reconsideration.
Round 2
Reviewer 1 Report
Comments and Suggestions for Authors
The authors have satisfactorily addressed all my concerns. Just a small observation. Please make sure you use decimal separators consistently. Specifically, replace commas with periods in “+123,5 m” and “36,9 s” so they read “+123.5 m” and “36.9 s,” respectively.
Author Response
REVIEWER 1
The authors have satisfactorily addressed all my concerns. Just a small observation. Please make sure you use decimal separators consistently. Specifically, replace commas with periods in “+123,5 m” and “36,9 s” so they read “+123.5 m” and “36.9 s,” respectively.
RESPONSE 1
Thank you for spotting this typo, which has been corrected
Reviewer 2 Report
Comments and Suggestions for Authors
Dear Authors,
Congratulations on all the new enhancements you have made to the manuscript. It is now much clearer, and the limitations of the study have been appropriately discussed, adding quality to the work. The conclusions accurately reflect the content and the findings. I have a few minor suggestions to further improve the quality of the manuscript:
Introduction
- It is necessary to include Feppon et al. (2025), reference 21 in the reference list, in the Introduction when presenting the device.
Methods
- The outcomes are well explained, but I recommend providing a bibliographic reference for each of them.
- It is necessary to provide a reference for this statement: “From a neurophysiological standpoint, spontaneous neuroplasticity and functional recovery potential are highest during the acute and subacute phases, making the latter an optimal period for evaluating rehabilitation interventions”.
- If you report the participants’ average age and standard deviation in the Results, you should also include them in the statistical analysis (descriptive variables).
Results
- Please clarify in the text what is meant by MCIDs or MDC/SRD (only the abbreviations are reported).
Discussion
- I recommend adding references to support this paragraph:
“ Direct comparisons between NeuroSkin® and existing peroneal-nerve FES devices (e.g., WalkAide, L300, ODFS) should be interpreted cautiously. Most available studies on FES for gait rehabilitation focus on single or dual-channel devices designed to stimulate the common peroneal nerve and correct isolated foot drop. In contrast, NeuroSkin® delivers synchronized, phase-specific stimulation to six lower-limb muscle groups using AI-based gait-phase detection, enabling broader neuromuscular engagement but inherently increasing system complexity.
While this difference may affect feasibility and usability, safety considerations are similar across FES modalities, as low-level electrical stimulation is well established”.
Author Response
REVIEWER 2
Dear Authors,
Congratulations on all the new enhancements you have made to the manuscript. It is now much clearer, and the limitations of the study have been appropriately discussed, adding quality to the work. The conclusions accurately reflect the content and the findings. I have a few minor suggestions to further improve the quality of the manuscript:
Introduction
- It is necessary to include Feppon et al. (2025), reference 21 in the reference list, in the Introduction when presenting the device.
RESPONSE
Thank you for pointing out this numbering mistake, Feppon et al. have been now properly cited (reference 19)
Methods
- The outcomes are well explained, but I recommend providing a bibliographic reference for each of them.
RESPONSE
Thanks for this suggestion, the following references have been added:
10MWT: [23] Watson, M. J. (2002). Refining the Ten-metre Walking Test for Use with Neurologically Impaired People. Physiotherapy, 88(7), 386–397. https://doi.org/10.1016/s0031-9406(05)61264-3
6MWT: [24] ATS Statement. Guidelines for the Six-Minute Walk Test (2002). American Journal of Respiratory and Critical Care Medicine, 166(1), 111–117. https://doi.org/10.1164/ajrccm.166.1.at1102
TUG: [25] Podsiadlo, D., & Richardson, S. (1991). The Timed “Up & Go”: A Test of Basic Functional Mobility for Frail Elderly Persons. Journal of the American Geriatrics Society, 39(2), 142–148. https://doi.org/10.1111/j.1532-5415.1991.tb01616.x
References [20] and [21] were already provided for SUS and NFAC
- It is necessary to provide a reference for this statement: “From a neurophysiological standpoint, spontaneous neuroplasticity and functional recovery potential are highest during the acute and subacute phases, making the latter an optimal period for evaluating rehabilitation interventions”.
RESPONSE : 
The sentence has been modified:
From a neurophysiological standpoint, spontaneous neuroplasticity and functional recovery potential are highest during the acute and early subacute phases—typically within the first three months after stroke—making the subacute period an optimal window for evaluating rehabilitation interventions. Clinical data support that a therapeutic sensitivity window of approximately 60–90 days post-stroke is associated with heightened treatment responsiveness, whereas interventions in the chronic phase (>6 months) exhibit markedly reduced efficacy [22]
And the following reference has been added: [22] Dromerick, A. W., Geed, S., Barth, J., Brady, K., Giannetti, M. L., Mitchell, A., Edwardson, M. A., Tan, M. T., Zhou, Y., Newport, E. L., & Edwards, D. F. (2021). Critical Period After Stroke Study (CPASS): A phase II clinical trial testing an optimal time for motor recovery after stroke in humans. Proceedings of the National Academy of Sciences, 118(39). https://doi.org/10.1073/pnas.2026676118
- If you report the participants’ average age and standard deviation in the Results, you should also include them in the statistical analysis (descriptive variables).
RESPONSE 
The following sentence has been added to the “Outcome measures and statistical analysis” part:
In addition, baseline descriptive variables (age, sex, and time since stroke) were summarized using descriptive statistics. Continuous variables are reported as mean ± standard deviation, and categorical variables as counts.
Results
- Please clarify in the text what is meant by MCIDs or MDC/SRD (only the abbreviations are reported).
The meanings of this acronyms have been introduced at the first occurrence in the text and also added to the abbreviations table
MCID Minimal Clinically Important Difference
MDC Minimal Detectable Change
SRD Smallest Real Change
Discussion
- I recommend adding references to support this paragraph:
“ Direct comparisons between NeuroSkin® and existing peroneal-nerve FES devices (e.g., WalkAide, L300, ODFS) should be interpreted cautiously. Most available studies on FES for gait rehabilitation focus on single or dual-channel devices designed to stimulate the common peroneal nerve and correct isolated foot drop. In contrast, NeuroSkin® delivers synchronized, phase-specific stimulation to six lower-limb muscle groups using AI-based gait-phase detection, enabling broader neuromuscular engagement but inherently increasing system complexity.
While this difference may affect feasibility and usability, safety considerations are similar across FES modalities, as low-level electrical stimulation is well established”.
RESPONSE 2
The paragraph has been updated with the following references:
Direct comparisons between NeuroSkin® and existing peroneal-nerve FES devices (e.g., WalkAide, L300, ODFS) should be interpreted cautiously. Most available studies on FES for gait rehabilitation focus on single or dual-channel devices that target the peroneal nerve to correct isolated foot drop and enhance gait speed in stroke populations [31,32]. In contrast, NeuroSkin® delivers synchronized, phase-specific stimulation to six lower-limb muscle groups, facilitating broader neuromuscular engagement at the cost of increased system complexity—a departure from singular muscle targeting strategies [33,34]. While this difference may affect feasibility and usability, and despite differences in configuration, safety considerations remain consistent across FES modalities, as the stimulation levels are similar and adverse events typically minor [35]. Finally, because this is a retrospective feasibility study without a control group, effectiveness comparisons with other devices would be premature. Future prospective trials should address comparative efficacy, usability, and workflow integration.
[31] Jaqueline da Cunha, M., Rech, K. D., Salazar, A. P., & Pagnussat, A. S. (2021). Functional electrical stimulation of the peroneal nerve improves post-stroke gait speed when combined with physiotherapy. A systematic review and meta-analysis. Annals of Physical and Rehabilitation Medicine, 64(1), 101388. https://doi.org/10.1016/j.rehab.2020.03.012
[32] Gil-Castillo, J., Alnajjar, F., Koutsou, A., Torricelli, D., & Moreno, J. C. (2020). Advances in neuroprosthetic management of foot drop: a review. Journal of NeuroEngineering and Rehabilitation, 17(1). https://doi.org/10.1186/s12984-020-00668-4
[33] Ferrante, S., Chia Bejarano, N., Ambrosini, E., Nardone, A., Turcato, A. M., Monticone, M., Ferrigno, G., & Pedrocchi, A. (2016). A Personalized Multi-Channel FES Controller Based on Muscle Synergies to Support Gait Rehabilitation after Stroke. Frontiers in Neuroscience, 10. https://doi.org/10.3389/fnins.2016.00425
[34] Berkelmans, S., Dominici, N., Afschrift, M., Bruijn, S., & Janssen, T. W. J. (2025). Feasibility and safety of automated multi-channel FES-assisted gait training in incomplete spinal cord injury. Journal of Rehabilitation Medicine, 57, jrm42638. https://doi.org/10.2340/jrm.v57.42638
[35] Khan, M. A., Fares, H., Ghayvat, H., Brunner, I. C., Puthusserypady, S., Razavi, B., Lansberg, M., Poon, A., & Meador, K. J. (2023). A systematic review on functional electrical stimulation based rehabilitation systems for upper limb post-stroke recovery. Frontiers in Neurology, 14. https://doi.org/10.3389/fneur.2023.1272992
Reviewer 3 Report
Comments and Suggestions for Authors
The manuscript was clearly improved, and it is now acceptable.
Author Response
The manuscript was clearly improved, and it is now acceptable.
Thank you very much
Reviewer 4 Report
Comments and Suggestions for Authors
Accept in present form.
Author Response
Accept in present form.
Thank you very much